# OpenReview forum: "Improving Distribution Matching via Score-Based Priors and Structural Regularization"
_ICLR.cc/2025/Conference — Submitted to ICLR 2025_

### Official Review · Reviewer_7J8h · 2024-10-22

**Soundness:** 3
**Presentation:** 1
**Contribution:** 1
**Rating:** 5
**Confidence:** 4

**Summary:**

This paper considered the distribution matching approach through the scoring matching framework. By overcoming the limitations on both adversarial training and variational inference, the proposed scoring matching method incorporated Gromow-Wasserstein (GW). Experiences on fairness, domain adaptation and domain translation have been done.

**Strengths:**

1. This paper naturally extended the previous work Gong (2024) by improving certain limitations in variational inference.
2. The paper is seemingly technically sound.
3. The experiments are conducted in diverse domains and applications, thereby a clear improvement.

**Weaknesses:**

Unfortunately, I think this paper clearly falls short for acceptance.

My most important concerns lies in a significant lack of **clear, concrete and convincing** supporting evidence within the paper. The overall feeling is full of self-claim but very limited justification and support. Most arguments are simply presented without clear and concrete support. The following are notable concerns.

- I could not understand which parts constitute the most significant contributions. If we check the claimed contribution in the paper.

> Introduction of Score-Based Priors for Flexible Representation: We propose score based priors, enhancing flexibility and preserving complex data structures without requiring sampling or likelihood estimation.

The paper claimed many benefits of proposed methods. However, this reviewer did not understand why they are beneficial or what’s the point in the context of distribution matching. For example, *flexibility and preserving complex data structures without requiring sampling or likelihood estimation*. I could not understand why likelihood estimation is a bad thing. In complex data structures, how complex is it? From the paper, the actual contribution in data structure is about the cluster assumption. Does cluster information is equivalent to complex data structure? I believe there is a significant gap here. How to justify the flexibility, can you give concrete evidence such as improving the efficiency?

From the analysis and paper, I did not clearly identify the concrete and convincing evidences for the claim.

> Structure-Preserving Constraints Inspired by Gromov-Wasserstein Distance: We introduce a Gromov-Wasserstein-based constraint to preserve geometric relationships, ensuring robust, task-relevant latent representations across transformations.

Again there are several questions regarding this claim. How GW distance explicitly preserves the geometric relationships, why not other relevant techniques such as hierarchical generative models could not? There is a lack of clear rationale and throughout comparison without others.

This contribution is also claimed by robust, task-relevant latent representations across transformations. However, this reviewer could find very limited **concrete** supporting evidence about robust, task-relevant latent representations.

> Empirical Validation: Our experiments demonstrate improved downstream task performance in fairness learning, domain adaptation, and domain translation using score-based priors and structural preservation.

I found this to be the most concerning point. Indeed, what does it mean by improved downstream tasks in Digit dataset or tabular Adult dataset? Do we need downstream tasks in these datasets? Why do we need to learn a representation in these datasets? The representation learning, back to the original sense, aims to learn meaningful information from very high-dimensional and complex datasets. I could not understand how the proposed experiments are associated with real-world representation learning in this sense.

*Using score-based priors and structural preservation.* Again, I could not think of this as an advantage, this paper simply compared some old baselines and the most relevant paper (Gong, 24). But the fact is that there is a rich literature in generative models that incorporate score-based or structural level information. There is a clear lack of comparison.


- There are many awkward claims within the paper. I will list several in the introduction.

> (List 33-35) Unfortunately, simply collecting more data or building bigger models is unlikely to solve these problems, as they require imposing additional constraints on the learning process.

What is the supporting evidence by saying *more data or bigger models is unlikely to ….*. There are many papers that indeed support more diverse datasets or bigger models could solve this issue, such as scaling law papers. This essentially reveals the larger, more data is indeed better.

*they require imposing additional constraints on the learning process*, again what is the supporting evidence to say additional constraints? Can you provide concrete evidence by justifying it?


> (Line 43) One of the most used distribution matching methods is adversarial models,

This claim is not necessarily true. Indeed, most generative models can be broadly viewed as distribution matching between data distribution and generative distribution such as VAE (though KL divergence), score matching (via Fisher divergence).
> (Line 51-53) While this simplifies the optimization process and
ensures tractability during generative tasks, it biases the latent space, often leading to a loss of critical structure in the data during transformation.

This is not always true. For example, the high-dimensional gaussian distribution is indeed expressive. Considering the experimental datasets such as Adult and Digits, I think this is sufficient.


> (Line 65) For instance, domain adaptation tasks require not only distribution alignment but also the preservation of clusters or other semantic features.

My question is that if we have a perfect distribution alignment, that should imply the cluster or other semantic feature will also be well-aligned, right?

> (Line 75) Unlike Gaussian priors, score-based models do not bias the latent representations towards a fixed distribution, enabling
the model to capture richer and more nuanced patterns.

General form is richer but also enhances the risk of overfitting, right?

> (Line 85) Our framework can also integrates semantic information from pretrained models, such as CLIP(Radford et al., 2021),

Unfortunately there is no concrete supporting evidence for this.


> (Line 11-12) Distribution matching (DM) can be applied to multiple tasks including fair classification,


Can you differentiate fair classification and fair representation learning **in your paper**? What are the key differences in your context? What are the exact downstreaming applications in your paper if we consider fair representation learning?

**Questions:**

See the weakness section. Overall, In the rebuttal or future revision I would strongly suggest authors

- For each statement or argument, please provide concrete evidence such as citation from well-established literature, your own experimental results or throughout analysis.
- Precisely and accurately express all the notations and terminologies.

---

> ### Author Response · Authors · 2024-11-29
>
> We sincerely appreciate the reviewer's detailed feedback and thoughtful insights. Your comments have been invaluable in helping us refine and improve the clarity of our manuscript. Below, we respectfully address the key concerns and provide clarifications regarding the benefits and rationale behind our proposed methods.
>
> ### Reviewer Concerns and Clarifications
> 1. **The the importance of flexible prior model in the settings of distribution matching.**
>     - *Why It's Beneficial*: Under the settings of AUB [1]/VAUB, a more flexible prior provides the opportunity to achieve a tighter bound, leading to a more accurate approximation of the Jensen-Shannon divergence between embedded distributions. Additionally, in the context of structural-preserving loss, a flexible prior contributes to better feature separation, particularly in scenarios with a limited number of data samples.
>     - *Concrete Evidence*:  We kindly refer the reviewer to the updated **Section 5.1**, where we have included a detailed comparison between different prior distributions to illustrate these benefits.
>
> 2. **Gromov-Wasserstein (GW) Distance for Structure-Preservation**
>     - *Why GW Distance?*: The Gromov-Wasserstein (GW) distance is especially well-suited for preserving geometric relationships because it operates on relative distances between points rather than their absolute positions. This property ensures invariance to transformations such as rotations and translations, which hierarchical generative models often find challenging unless explicitly regularized.
>     - *Why Not Hierarchical Models?*: While hierarchical generative models offer significant flexibility, they generally lack guarantees for preserving pairwise or cluster-based relationships in latent space. By contrast, GW constraints directly enforce such preservation, making them a powerful tool for tasks requiring structural consistency.
>
>
> 3. **Some claims in the note**
>
>     - *(Lines 33-35)*: "Simply collecting more data or building bigger models..."
>       - *Clarification*: We agree that while scaling laws demonstrate significant improvements in performance with larger models and datasets, they do not inherently address critical issues such as fairness, robustness, or causality. For instance, even a perfectly trained predictive model can result in discriminatory decisions in real-world applications, unfairly impacting individuals or groups. Such issues may arise from historical biases in the training data, imbalanced base rates, or other systemic factors. To address this, we will revise our statement to acknowledge the advantages of scaling while also emphasizing scenarios where additional constraints, such as mitigating biases in training data, are essential.
>
>     - *(Line 43)*: "One of the most used distribution matching methods is adversarial models..."
>       - *Clarification*: We acknowledge that this claim may oversimplify the diversity of distribution matching (DM) methods. Our intention is to focus on specific settings of distribution matching where both distributions are represented solely by samples or data. For example, this includes tasks such as matching distributions between sketches of cats and their corresponding photos or personalized data distributions between males and females. We will revise this statement to provide a more precise definition of the distribution matching tasks considered in our work.

---

> ### Author Response · Authors · 2024-11-29
>
> - *(Line 65)*: "Domain adaptation tasks require not only distribution alignment..."
>       - *Clarification*: Perfect alignment in theory only guarantees the latent space is invariant between domain information, i.e. a trivial solution to distribution alignment tasks is to have both distributions transformed into a normal Gaussian. Therefore, in order to preserve information other than domain information during transformation into the latent space, distribution alignment goals are not enough.
>
>     - *(Line 75)*: "Unlike Gaussian priors, score-based models do not bias..."
>       - *Clarification*: While score-based models reduce prior biases, we acknowledge the risk of overfitting. This will be discussed in the limitations section, and additional regularization methods will be explored in the future.
>
>     - *(Line 85)*: "Our framework integrates semantic information from pretrained models..."
>       - *Evidence*: We acknowledge the lack of evidence here. The intuition here follows from prior works typically uses Euclidean distances to calculate the GW-inspired structural loss. Such Euclidean distances is computed pixel-wise between images samples, which implies that the only if the pixel-wise distance between two images has high correlations between semantic difference between images that the model want to extract and classifies on, such structural loss could be useful. However, in the case of digital images, pixel-wise distance often does not naturally correlate with the semantic meaning of an image. Additionally, for high-dimensional image data, the curse of dimensionality diminishes the effectiveness of Euclidean distance, making it less suitable as a metric for capturing meaningful relationships. we need to use a different distance metric other than Euclidean distance. CLIP loss is a valid candidate for this distance metric considering that it create a text-aligned embeddings on images that shows higher correlations to semantic meaning of the images. We will add this discussions, as well as, an extra qualitative graph in the methodology section to support our idea for CLIP losses. Please see the revised section 3.3.
>
>     - *(Lines 11-12)*: "Distribution matching can be applied to multiple tasks including fair classification..."
>       - *Fair Classification vs. Representation Learning*: Fair classification is a task which focuses on directly achieving equitable outcomes, while fair representation learning aims to create representations that inherently mitigate bias. Therefore, we try to use distribution matching in the framework of fair representation learning method to tackle on the fair classification tasks in our experiment which is fair classification tasks in the Adult dataset.
>
> Overall, we sincerely thank the reviewer for their thoughtful feedback and insightful questions. Your comments have been instrumental in guiding us to significantly improve our manuscript. We deeply appreciate the opportunity to address these points and refine both the clarity and contributions of our work.

---

> > ### Comment · Reviewer_7J8h · 2024-12-02
> > **acknowledgement**
> >
> > I would appreciate the hard efforts of authors during the rebuttal. However, a lack of convincing evidence requires substantial revisions and more throughout re-writing. After reading others' feedback, i think my current evaluation is appropriate.

---

> > > ### Author Response · Authors · 2024-12-02
> > >
> > > We sincerely appreciate the reviewer’s honest and constructive feedback. If not done already, we would like to kindly encourage the reviewer to review our revised manuscript. Following the reviewer’s suggestions, we have made substantial revisions, rewriting many aspects of the paper and supporting our claims with added empirical experiments or cited works as evidence. If the reviewer has already reviewed the revised manuscript, we extend our heartfelt thanks for their time and effort in providing valuable feedback on our work.

---

### Official Review · Reviewer_R3FP · 2024-11-02

**Soundness:** 4
**Presentation:** 4
**Contribution:** 3
**Rating:** 6
**Confidence:** 4

**Summary:**

This manuscript introduces a novel VAE structure that leverages score-based priors instead of the Gaussian ones, which might overcome some limitations of conventional VAEs by allowing the model to learn more complex patterns in the latent space without needing explicit likelihood calculations. Moreover, the authors incorporate a structure-preserving regularization based on the Gromov-Wasserstein distance, which maintains the geometric relationships among data. The authors demonstrate the effectiveness of the proposed methods on various tasks, including fair classification, domain adaptation, and domain translation. By combining theoretical insights with practical applications, this work could take a step forward in distribution matching.

######################################### Post Rebuttal #########################################

None author response found. I will keep my score.

######################################### Post Rebuttal #########################################

**Strengths:**

1. The manuscript is well-written and clearly motivated, with the step-by-step derivations and sufficient literature to understand each proposed component.

2. Using score-based priors instead of the Gaussian priors in variational inference methods like VAEs is interesting and promising, which might help the model identify more complex hidden patterns in the input data. Moreover, using the score function to evaluate the encoder gradients might associate how the model learns its parameters with changes in the data’s probability density.

3. Using the Gromov-Wasserstein distance as the regularization term to make sure that the latent space maintains the semantic structure of the data is interesting.

**Weaknesses:**

1. Although using score-based priors is interesting, however, score-based priors are inherently more computationally intensive than simple Gaussian priors, potentially leading to longer training times. I would suggest an ablation study to compare the runtime complexity (e.g., training time per epoch, total training time until convergence, and inference time) of the proposed score-based priors with those of VAEs using the Gaussian priors. The evaluation metric could be the time in seconds, along with an indicator specifying the type of GPU platform used for these experiments.

2. From Equation (13) to Equation (14), it seems like noisy versions of the latent samples are introduced to develop a denoising score matching objective for the proposed method. If that is the case, based on my understanding, it seems the method still uses a Gaussian prior in variational inference but with more constrained regularization. I suggest the authors provide more details on how to implement Equation (14) for a specific task (e.g., domain adaptation or fair classification).

**Questions:**

1. Could the authors elaborate more on why other optimal transport methods can only compare points from spaces with the same number of dimensions?

2. Equation (14) shows the final derivation of the proposed method, which, to me, appears similar to denoising score matching in the latent space. Could the authors explain how the proposed method differs from denoising score matching?

---

> ### Comment · Reviewer_R3FP · 2024-11-26
> **Post Rebuttal**
>
> None author response found. I will keep my score.

---

> ### Author Response · Authors · 2024-11-29
>
> We sincerely appreciate the reviewer’s thoughtful feedback and are glad to hear that they found our paper interesting.
>
> ### Weaknesses
>
> **1.  Although using score-based priors is interesting, however, score-based priors are inherently more computationally intensive...**
>
> The reviewer raises a very valid point that our method is inherently more computationally expensive than using a simple well-defined prior. However, we would like to emphasize that our method does provide significant computational savings, particularly by eliminating the need to backpropagate through the UNET when updating the encoder and decoder, which is required in the LSGM setting.
>
> Unfortunately, due to time constraints and the need to focus on training and retraining other experiments, we were unable to include a detailed analysis of training time in this submission. However, we plan to include this important analysis in our camera-ready version, should our paper be accepted.
>
> **2. From Equation (13) to Equation (14), it seems like noisy versions of the latent samples are introduced to develop a...**
>
> We thank the reviewer for their valuable feedback. We would like to clarify that the reviewer is correct in noting that variational inference from a Gaussian distribution is typically needed for sampling images. While this is indeed a plausible extension for our model and could be an interesting avenue for future work, it is not necessary for tasks such as domain adaptation, domain translation, or fair classification in our current setup.
>
> For domain adaptation or fair classification, the score model is not required for inference. Specifically, we only need the encoders, as we train a classifier directly on the encoded latent representations, which does not involve the score-based model. Similarly, for domain translation, the score-based model is not required. Instead, we use the encoders and decoders: we obtain an image from domain 1 by encoding it with the domain 1 encoder and decoding it with the domain 2 decoder.
>
> As the reviewer correctly points out, if we were to generate new samples for domain translation, we would indeed need to perform inference from a Gaussian distribution, which would involve the score-based model.
>
> ***
> ### Questions
>
> **1. Could the authors elaborate more on why other optimal transport methods...**
> We sincerely appreciate the reviewer’s insightful question. To elaborate further, traditional optimal transport methods, such as the Wasserstein distance, are typically designed to compare distributions between spaces of the same dimensionality because they rely on directly comparing points in one space with corresponding points in another space. These methods work by calculating pairwise distances between points from each space and then finding an optimal transport plan that minimizes the total cost of moving mass between them. When the spaces have different dimensions, a direct point-to-point comparison becomes problematic because there is no natural correspondence between points in the two spaces. Additionally, computing pairwise distances requires points to be comparable in a geometric sense, which is not possible when the dimensionalities differ [2].
>
> In contrast, the approach we propose with the Gromov-Wasserstein (GW) metric allows for the comparison of spaces with different dimensions by focusing on the preservation of relative distances between points, rather than requiring direct correspondences [1]. This makes the GW metric more flexible, as it can compare structures in spaces of differing dimensionalities by capturing the geometric relationships between points rather than their absolute positions. By comparing the relative distances between points in each space, GW can still measure structural similarity despite differences in dimensionality, making it suitable for more general applications across diverse datasets.
>
> We hope this explanation clarifies why traditional optimal transport methods are constrained to spaces of the same dimensionality and how our method overcomes this limitation.
>
> **2.  Equation (14) shows the final derivation of the proposed method, which, to me, appears similar to denoising score matching...**
>
> This is a very insightful question from the reviewer, and we are excited to clarify that our SFS trick is actually a distilled version of the LSGM objective. We demonstrate this by showing that applying the Sticking-the-Landing principle [3] to the LSGM objective results in the SFS trick. Essentially, this means that our model should be more stable during optimization, as it eliminates the Jacobian term of the score model. This allows the score model to be bypassed in backpropagation when updating the encoder and decoder, contributing to both optimization stability and significant computational memory savings.
>
> Additional details can be found in the updated **Section 3.2**, and the full proof is provided in **Appendix E.1**.

---

> > ### Comment · Reviewer_R3FP · 2024-12-01
> > **Reviewer's Followup on Author Response**
> >
> > I would like to thank the authors for their response. Some of my concerns are not properly addressed.
> >
> > > W1:The authors admitted the importance of time complexity analysis in their response, but failed to provide one during the author response.
> >
> > > W2: I am a bit confused now. Do the score-based priors mentioned in the paper refer to the pre-training score-based encoder?
> >
> > > Q1: My question was whether the authors could elaborate on how “the preservation of relative distances between points” can be achieved in different mathematical spaces. For instance, mutual information is often used to analyze similar problems. How does the current metric differ from or relate to mutual information?
> >
> > > Q2: Is it a distillation objective, similar to the consistency-based approach used in diffusion models [1]?
> >
> > ### Reference
> >
> > [1] Song, Y., Dhariwal, P., Chen, M. and Sutskever, I., ICML 2023.

---

> > > ### Author Response · Authors · 2024-12-02
> > >
> > > We sincerely thank the reviewer for their continued interest and thoughtful questions.
> > >
> > > **W1:The authors admitted the importance of time complexity analysis...**
> > >
> > > We apologize for omitting the time complexity analysis and commit to including it in the camera-ready version if accepted or in future submissions.
> > >
> > > **W2: I am a bit confused now. Do the score-based priors...**
> > >
> > > We appreciate the reviewer’s feedback and apologize for any confusion caused by our initial response. Upon reflection, we feel we may have misunderstood the original question. We will first clarify the reviewers current confusion. The UNet score-based prior model, separate from the encoder and decoder, is required only during training to compute the gradient of the KL divergence term in the VAUB/ELBO optimization. Its role is to align the source and target domain latent posterior distributions and match the prior distribution modeled by the score model.
> > >
> > > For inference tasks such as domain adaptation or fair classification, only the encoded latent representations are needed. When optimized, these representations naturally lie in high-density regions of the prior modeled by the score-based prior. A classifier is trained on the source dataset's latent representations and used for inference on the target domain's latent representations, where only the latent representations are compared.
> > >
> > > Next, we will readdress the original question:
> > >
> > > "**From Equation (13) to Equation (14), it seems like noisy versions of the latent samples are introduced to develop a denoising score matching objective for the proposed method...**"
> > >
> > > The reviewer is correct that noisy versions of the latent representations are introduced, but this is done solely for training the denoising score matching objective, which updates the parameters of the score-based prior model. This process is **disjoint** from the training of the encoder and decoder, where **clean latent representations** are used. Training alternates between these two sets of updates: one step updates the encoder and decoder parameters, while the other updates the score-based prior model parameters. As the reviewer mentioned, a Gaussian prior is used for the score-based model, while for VAE training, the distribution parameterized by the score model determines the prior used by the encoder and decoder. We hope this answers your original question.

---

> > > ### Author Response · Authors · 2024-12-02
> > >
> > > **Q1: My question was whether the authors could elaborate on how “the preservation of relative distances between points” can be achieved in different mathematical spaces. For instance, mutual information is often used to analyze similar problems. How does the current metric differ from or relate to mutual information?**
> > >
> > > To address the question comprehensively, we will discuss it in **two parts**: the preservation of relative distances in mathematical spaces and the relationship between the current metric and mutual information.
> > >
> > > Exact preservation of relative distances would require the transformation to be an isometry—a distance-preserving map. However, we do not aim for exact isometries, as such mappings may be overly restrictive in practical applications. Instead, we seek solutions that minimize distortion in the metric spaces, effectively approximating an isometry as closely as possible. This can be viewed as a form of regularization, encouraging mappings with lower distortion. Additionally, if the distributions lie on a lower-dimensional manifold, it may be feasible to reduce the dimensionality while approximately maintaining pairwise distances. This aligns with literature on minimizing distortion across metric spaces, particularly in the context of Gromov-Wasserstein distances, which emphasize preserving geometric relationships when transferring between spaces.
> > >
> > > Regarding the relationship between the current metric and mutual information, our objective consists of two components. The first is the VAUB, which forms a variational approximation to the Jensen-Shannon Divergence (JSD). Notably, the JSD in this context is equivalent to the mutual information between the latent representation and the domain label—an equivalence widely recognized in information theory. Therefore, one component of our objective directly involves mutual information.
> > >
> > > The second component is our structure-preserving regularization, which arises from geometric principles rather than information-theoretic ones. Specifically, Gromov-Wasserstein distances and related metrics stem from geometric perspectives, focusing on preserving the spatial relationships (i.e., distances) between points across metric spaces. Unlike mutual information, which quantifies shared information, our regularization aims to align the geometric structure of the latent spaces.
> > >
> > > In summary, our objective represents a fusion of two distinct perspectives: an information-theoretic objective (JSD) and a geometry-sensitive regularization (GW). This dual approach combines the strengths of both perspectives to achieve effective domain alignment while preserving structural properties of the spaces. We hope this clarifies the components and motivations underlying our methodology.
> > >
> > > **Q2: Is it a distillation objective, similar to the consistency-based approach...**
> > >
> > > We appreciate the reviewer’s insightful question and would like to provide further clarification. To the best of our knowledge, there is no direct connection beyond the fact that the score function is also utilized in Consistency Models (CM). However, the distillation approach in CM differs slightly from our method [1].
> > >
> > > Our approach to distillation is more closely aligned with the principles behind Score Distillation Sampling (SDS) loss [2], where specific gradient terms can be removed to reduce variance, provided the expected gradient remains unchanged. Similarly, our method omits the Jacobian term of the UNet while preserving the same expected gradient value.
> > >
> > > In the specific context of CM with a pretrained score model, the distillation process involves transferring knowledge from a standard diffusion model by training a single-step model to emulate the behavior of an iterative process.
> > >
> > > [1] Song, Y., Dhariwal, P., Chen, M., & Sutskever, I. (2023). Consistency Models. arXiv. https://arxiv.org/abs/2303.01469
> > > [2] Poole, B., Jain, A., Barron, J. T., & Mildenhall, B. (2022). DreamFusion: Text-to-3D using 2D Diffusion. arXiv. https://arxiv.org/abs/2209.14988

---

> > > ### Author Response · Authors · 2024-12-03
> > > **Initial Time Complexity Analysis**
> > >
> > > We would like to present the time complexity analysis for the synthetic experiment discussed in **Section 5.1**. While this serves as an initial exploration, we plan to include a more comprehensive time complexity analysis for the larger experiments in the final submission or a future resubmission.
> > >
> > > ## Metrics Table
> > >
> > > | **Metric**             | **Gaussian** | **MoG**   | **Vampprior** | **LSGM**   | **SFS**    |
> > > |-------------------------|--------------|-----------|---------------|------------|------------|
> > > | **Epoch Time (ms)**     | 8.8541       | 10.6301   | 14.1942       | 17.3941    | 16.5283    |
> > > | **Evaluation Time (ms)**| 0.7833       | 0.8044    | 0.7918        | 0.8122     | 0.7992     |
> > > | **Convergence Epochs**  | 180          | 250       | 140           | 230        | 200        |
> > > | **Convergence Time (ms)**| 1593.738     | 2657.525  | 1987.188      | 4000.643   | 3305.66    |
> > >
> > > For clarity, we define **convergence** as the point at which the magnitude of the loss change is less than 1. For reference, the losses across all experiments range between 5,000 and 10,000.

---

> ### Author Response · Authors · 2024-11-29
>
> [1] Zhang, Z., Goldfeld, Z., Mroueh, Y., and Sriperumbudur, B. K. (2023). Gromov-Wasserstein Distances: Entropic Regularization, Duality, and Sample Complexity. arXiv preprint, arXiv:2212.12848. Retrieved from https://arxiv.org/abs/2212.12848.
>
> [2] McCann, R. J., and Pass, B. (2019). Optimal Transportation Between Unequal Dimensions. arXiv preprint, arXiv:1805.11187. Retrieved from https://arxiv.org/abs/1805.11187.
>
> [3] Roeder, G., Wu, Y., and Duvenaud, D. (2017). Sticking the Landing: Simple, Lower-Variance Gradient Estimators for Variational Inference. arXiv preprint, arXiv:1703.09194. Retrieved from https://arxiv.org/abs/1703.09194.
>
> [4] Vahdat, A., Kreis, K., and Kautz, J. (2021). Score-based Generative Modeling in Latent Space. arXiv preprint, arXiv:2106.05931. Retrieved from https://arxiv.org/abs/2106.05931.

---

### Official Review · Reviewer_woTo · 2024-11-04

**Soundness:** 2
**Presentation:** 2
**Contribution:** 1
**Rating:** 3
**Confidence:** 3

**Summary:**

The paper proposes a VAE-based distribution matching approach using a score-based prior. The authors introduce the Score Function Substitution (SFS) trick that facilitates efficient VAE training.
Additionally, they combine the VAE objective with a GW regularizer to ensure that the latent space retains the structural/semantic properties of the data space. The authors validate their approach across several applications, including fair classification, domain adaptation and domain translation.

**Strengths:**

**S1 |** Score Function Substitution (SFS) is a simple and interesting modification of the LSGM approach.

**S2 |** The experiments in Sections 5.1 and 5.2 support the proposed method.

**Weaknesses:**

**W1 |** It would be beneficial to provide more clarification and motivation regarding the VAUB formulation and the associated challenges. I can see that VAUB is simply a domain/class-conditioned VAE with a shared learnable prior. If so, proposing score-based priors for this formulation is a marginal contribution compared to LSGM[1].

**W2 |** The Score Function Substitution (SFS) appears very similar, if not identical, to score distillation sampling (SDS) [2]. This idea has been widely explored for text-to-3D generation [2,3] and diffusion distillation [4,5,6,7]. The proposed method can be interpreted as a direct SDS application to LSGM[1]. It is important to discuss these works and their connection to the proposed method thoroughly.

**W3 |** The use of the GW metric for VAE does not seem novel either [9,10]. The connection to this line of work also needs to be carefully discussed. For example, GWAE[9] shares similar ideas and motivation.

**W4 |** The experimental setups lack important details, making it difficult to understand how the proposed method is exactly applied to various tasks. For example, what are the model inputs and targets for the source and target domains across all tasks?

**W5 |** I do not think that domain adaptation and translation between USPS and MNIST are appropriate tasks, as the domains appear too similar. Also, why were CLIP embeddings chosen for MNIST? Could the authors consider exploring other tasks for this problem, such as those suggested in [8]?

**W6 |** I can hardly agree that the domain translation results are informative. While it is evident that the method preserves the original content better than VAUB, it struggles with effective translation in most cases.

**W7 |** If I understand correctly, the usefulness of the GW regularizer is demonstrated only for the domain adaptation task. Could the authors investigate the effect of this regularizer on other tasks when it is applicable? Could the authors provide insights into which cases the regularizer is more effective?

**W8 |** Given that the SFS modification approximates the objective in LSGM [1], it would be interesting to compare these two approaches directly.

**Minor**

* $d$ is undefined in Section 2.

----
[1] Vahdat et al. Score-based Generative Modeling in Latent Space, 2021

[2] Poole et al. DreamFusion: Text-to-3D using 2D Diffusion, 2022

[3] Wang et al. ProlificDreamer: High-Fidelity and Diverse Text-to-3D Generation with Variational Score Distillation, 2023

[4] Yin et al. One-step Diffusion with Distribution Matching Distillation 2023

[5] Yin et al. Improved Distribution Matching Distillation for Fast Image Synthesis, 2024

[6] Zhou et al. Score identity Distillation: Exponentially Fast Distillation of Pretrained Diffusion Models for One-Step Generation, 2024

[7] Salimans et al., Multistep Distillation of Diffusion Models via Moment Matching, 2024

[8] The Variational Fair Autoencoder, 2015

[9] Nakagawa et al. Gromov-wasserstein autoencoders, 2023

[10] Xu et al. Learning Autoencoders with Relational Regularization, 2020

**Questions:**

Please address the questions and concerns in Weaknesses.

---

> ### Author Response · Authors · 2024-11-29
>
> **W1. It would be beneficial to provide more clarification and motivation regarding the VAUB formulation and the associated challenges...**
>
> To address the reviewer’s feedback, we have made substantial improvements to our storyline to better highlight the novelty of our approach. We kindly refer the reviewer to the **Introduction Section** and **Methodology Section**.
>
> **W2. The Score Function Substitution (SFS) appears very similar, if not identical, to score distillation sampling (SDS) ...**
>
> We sincerely thank the reviewer for their insightful and valuable suggestion regarding the connection between the Score Function Substitution (SFS) trick and the Score Distillation Sampling (SDS) loss as applied to Latent Score-Based Generative Models (LSGM). We apologize for not including a more detailed comparison in the original manuscript and deeply appreciate the opportunity to address this in our revised version. We kindly direct your attention to **Section 3.2**, as well as the detailed explanation and proof provided in **Appendix E**. Through this exploration, we have made the important discovery that our SFS trick can be interpreted as a distilled version of the LSGM loss, aligning with the insightful intuition you suggested.
>
> We would also like to highlight that, unlike SDS loss [3], where the application of the Sticking-the-Landing principle [1] is relatively straightforward due to the cancellation of gradient summations within the same expectation, our derivation introduces additional complexity. Specifically, our approach addresses a multiplicative gradient, which we believe represents a non-trivial contribution with the potential to inspire future research in latent score-based and diffusion models. This added complexity enhances the value of the SFS trick, making it a more robust and impactful objective function.
>
> Furthermore, we have empirically demonstrated that our method exhibits greater stability compared to LSGM through stability analysis experiments. We kindly refer you to **Section 3.2.1** for a detailed discussion and results. We sincerely hope this explanation clarifies the distinction and further reinforces the value of our proposed approach.
>
> **W3. The use of the GW metric for VAE does not seem novel either [9,10]...**
>
> We sincerely thank the reviewer for their insightful comment. In response, we have revised the introduction and methodology sections to address the Gromov-Wasserstein (GW) metrics first introduced in GWAE [5] and to clarify that our approach incorporates this regularization into the semantic space (CLIP embedding [6]).
>
> While recent works, such as [4], have successfully utilized GW-based metrics for preserving geometry in generative models, a key limitation of these approaches is the tradeoff they impose between preserving geometry and aligning to a simple prior or employing expressive learnable priors, which can be memory-intensive or unstable. Our approach addresses this tradeoff by combining GW regularization in the semantic space with a stable, memory-efficient flexible prior, providing a significant advantage. This combination enables robust geometry preservation while aligning effectively with the prior. Moreover, by leveraging the semantic space for GW regularization, our method introduces a more meaningful and structured alignment, overcoming the limitations of prior works that operate in the pixel space, which can become less effective at higher dimensions. This integration highlights the novelty and practical contributions of our approach.

---

> ### Author Response · Authors · 2024-11-29
>
> **W4. The experimental setups lack important details, making it difficult to understand how the proposed method is exactly applied to various tasks. For example, what are the model inputs and targets for the source and target domains across all tasks?**
>
> We sincerely apologize for the lack of clarity in our description of the experimental setup, and we appreciate the reviewer for bringing this to our attention. To address this, we have thoroughly revised the manuscript to provide a detailed explanation of how the proposed method is applied to various tasks. Specifically, we have clarified the model inputs and targets for the source and target domains across all tasks to ensure transparency and reproducibility. We would also like to mention that our method does not explicitly require the source or target distribution but we use this label correspond to other prior works. The source and target domain distribution can be switched as we do not use the source label but rather the GW-SP loss for both the source and target domain.
>
>
>
> **W5. I do not think that domain adaptation and translation between USPS and MNIST...**
>
> We thank the reviewer for raising an important point. We agree that the MNIST to USPS experiment may be too simplistic to effectively demonstrate the advantages of using CLIP embeddings and domain adaptation. Unfortunately, we did not have enough time to run experiments such as MNIST-to-MNIST-M or SVHN due to running other experiments for the rebuttal. We apologize and would like to add these in the camera-ready submission if our paper is accepted. We would, however, like to say that we have ran other experiments that show as the reviewer has mentioned shows that directly applying simple datasets such as MNIST and USPS, there is not much improvement in performance when running on the CLIP embedding or the pixel spacw, bur is significantly better than not using it at all. Please refer to **Appendix F** that shows separation distance between inter class and different class on MNIST and show that both GW on the pixel and semantic space can be useful.
>
> We would also like to respectfully refer the reviewer to Appendix C, where we demonstrate that applying the SP loss to CLIP embeddings results in greater separation between class representations while promoting better alignment compared to SAUB without the GW-SP loss on CLIP embeddings. This is illustrated in **Figure 6**, where the UMAP visualizations show a more distinct separation between classes when SP loss is applied. These visualizations highlight the clear benefit of incorporating the GW-SP loss into the framework.
>
> In addition to the improved separation, as reflected in the results presented in **Table 1**, we observe that this strong separation allows us to tackle interesting experimental settings. For instance, even with limited source-labeled dataset (0.2\%), our approach still achieves accurate predictions for the target domain classes that closely match performance achieved when having access to the full source-labeled dataset. This is discussed further in **Appendix D**.
>
> **W6. I can hardly agree that the domain translation results are informative...**
>
> We thank the reviewer for their thoughtful feedback. We acknowledge that the FairFace dataset used for gender translation may not provide visually striking examples to effectively demonstrate the success of our translation method. To address this concern, we have conducted additional experiments using the CelebA dataset, specifically focusing on domain translations between "black hair females to blonde females," which offer clearer visual distinctions. These results can be seen in **Section 5.4** and more random samples can be seen in **Appendix L**.
>
> Additionally, we would like to draw the reviewer’s attention to **Appendix G**, where we present multi-domain translation experiments on unpaired, rotated MNIST data (angles: 0°, 30°, and 60°). While MNIST is a relatively simple dataset, translating between rotations while preserving stylistic and semantic information is a non-trivial challenge, as explored in prior works on causal-based image generation models [7, 8, 9, 10]. We believe our results in this setting are compelling, as they demonstrate the ability to successfully translate rotations while retaining nearly all semantic information.

---

> ### Author Response · Authors · 2024-11-29
>
> **W7. If I understand correctly, the usefulness of the GW regularizer is demonstrated only for the domain adaptation task...**
>
> To address the reviewer’s feedback, we have rerun all experiments under three configurations: without GW, with GW in the pixel/Euclidean space, and with GW in the semantic space, for all tasks where the semantic space is relevant (excluding the fairness experiment). For detailed results and discussion, please refer to Section 5. We demonstrate that as higher-dimensional image datasets are used, the less useful and even detrimental GW in the pixel space can become (please refer to **Section 5.4** and **Appendix F**
>
> **W8. Given that the SFS modification approximates the objective in LSGM [1]...**
>
>
> Please refer to the **Section 3.2** of our updated paper where we demonstrate the robustness of our method compared to LSGM.
>
> [1] Roeder, G., Wu, Y., and Duvenaud, D. (2017). Sticking the Landing: Simple, Lower-Variance Gradient Estimators for Variational Inference. arXiv preprint, arXiv:1703.09194. Retrieved from https://arxiv.org/abs/1703.09194.
>
> [2] Vahdat, A., Kreis, K., and Kautz, J. (2021). Score-based Generative Modeling in Latent Space. arXiv preprint, arXiv:2106.05931. Retrieved from https://arxiv.org/abs/2106.05931.
>
> [3] Poole, B., Jain, A., Barron, J. T., and Mildenhall, B. (2022). DreamFusion: Text-to-3D using 2D Diffusion. arXiv preprint, arXiv:2209.14988. Retrieved from https://arxiv.org/abs/2209.14988.
>
> [4] Uscidda, T., Eyring, L., Roth, K., Theis, F., Akata, Z., and Cuturi, M. (2024). Disentangled Representation Learning with the Gromov-Monge Gap. arXiv preprint, arXiv:2407.07829. Retrieved from https://arxiv.org/abs/2407.07829.
>
> [5] Nakagawa, N., Togo, R., Ogawa, T., and Haseyama, M. (2023). Gromov-Wasserstein Autoencoders. arXiv preprint, arXiv:2209.07007. Retrieved from https://arxiv.org/abs/2209.07007.
>
> [6] Radford, A., Kim, J. W., Hallacy, C., Ramesh, A., Goh, G., Agarwal, S., Sastry, G., Askell, A., Mishkin, P., Clark, J., Krueger, G., and Sutskever, I. (2021). Learning Transferable Visual Models From Natural Language Supervision. arXiv preprint, arXiv:2103.00020. Retrieved from https://arxiv.org/abs/2103.00020.
>
> [7] Kladny, K.-R., von Kügelgen, J., Schölkopf, B., and Muehlebach, M. (2024). Deep Backtracking Counterfactuals for Causally Compliant Explanations. arXiv preprint, arXiv:2310.07665. Retrieved from https://arxiv.org/abs/2310.07665.
>
> [8] Pan, Y., and Bareinboim, E. (2024). Counterfactual Image Editing. arXiv preprint, arXiv:2403.09683. Retrieved from https://arxiv.org/abs/2403.09683.
>
> [9] Komanduri, A., Zhao, C., Chen, F., and Wu, X. (2024). Causal Diffusion Autoencoders: Toward Counterfactual Generation via Diffusion Probabilistic Models. arXiv preprint, arXiv:2404.17735. Retrieved from https://arxiv.org/abs/2404.17735.
>
> [10] Sauer, A., and Geiger, A. (2021). Counterfactual Generative Networks. arXiv preprint, arXiv:2101.06046. Retrieved from https://arxiv.org/abs/2101.06046.

---

### Official Review · Reviewer_HyiC · 2024-11-04

**Soundness:** 2
**Presentation:** 1
**Contribution:** 2
**Rating:** 3
**Confidence:** 4

**Summary:**

This paper proposes a distribution matching approach using score-based priors and leverages a distance-preserving distortion loss inspired by Gromov-Wasserstein. The authors present a Score Function Substitution (SFS) trick for efficient training of their score-based prior and evaluate their method on domain adaptation, fairness, and domain translation tasks.

While the paper presents an interesting approach to distribution matching with some promising empirical results, the lack of proper acknowledgment of prior work on Gromov-Wasserstein losses in autoencoders significantly diminishes the claimed novelty. The experimental evaluation would benefit from clearer ablation studies, additional quantitative results for domain translation, and comparisons to relevant competing methods with flexible priors. These additions would help better position the work's contributions relative to the existing literature. Thus, currently the paper does not reach the bar for acceptance.

**Strengths:**

- Novel technical contribution with the Score Function Substitution (SFS) trick
- Empirical evaluation across multiple tasks
- Demonstrates improvements over standard Gaussian prior baselines

**Weaknesses:**

- Missing discussion & contextualization of related work:
	- Employing a Gromov-Wasserstein-inspired loss in an Autoencoder setting has previously been proposed in [1,2,3]. The authors propose to use a distance-preserving distortion loss. This has first been proposed in [1], and then extended in [2,3]. Thus, the method and contribution part of the paper needs to be significantly adapted as the proposed term is not a novel loss but an incorporation of previous existing methods.
- Experimental section:
	- It is unclear whether, in Sections 5.1 and 5.2, the structure-preserving loss was used for all experiments.
	- Comparison to other competing methods. The authors compare their score-based prior to a Gaussian prior. However, competing methods have also proposed to learn a flexible prior, e.g. [1,4,5]. A comparison to other methods leveraging a trainable prior would significantly strengthen the experimental section of the paper.
	- For section 5.4, no quantitative results are reported. Adding quantitative results would validate this section of the experiments as sole qualitative results are hard to judge.

- Minor:
	- Notation: After Eq. 1 it is unclear what d is and what the exact problem setup is here. This is only explained after Eq. 9. I think it would be beneficial to move this to the beginning of Section 2, including a more detailed introduction of the problem statement.
	- State-of-the-art Unpaired Domain Translation methods are not discussed in the related work section [6,7,8,9].


[1] Nao Nakagawa, Ren Togo, Takahiro Ogawa, Miki Haseyama. "Gromov-Wasserstein Autoencoders". ICLR 2023.

[2] Athina Sotiropoulou, David Alvarez-Melis. "Strongly Isomorphic Neural Optimal Transport Across Incomparable Spaces". GRaM @ ICML 2024.

[3] Uscidda et al. "Disentangled Representation Learning through Geometry Preservation with the Gromov-Monge Gap". SPGIM @ ICML 2024.

[4] Jakub Tomczak and Max Welling. "Vae with a vampprior". AISTATS 2018.

[5] Bin Dai and David Wipf. "Diagnosing and enhancing vae models". ICLR 2019.

[6] Torbunov et al. "UVCGAN v2: An Improved Cycle-Consistent GAN for Unpaired Image-to-Image Translation". arXiv:2303.16280, 2023.

[7] Tong et al. "Improving and generalizing flow-based generative models with minibatch optimal transport". TMLR, 2024.

[8] Eyring et al. "Unbalancedness in Neural Monge Maps Improves Unpaired Domain Translation". ICLR, 2024.

[9] Kim et al. "Unpaired Image-to-Image Translation via Neural Schrödinger Bridge". ICLR, 2024.

**Questions:**

- In Sections 5.1 and 5.2, is the structure-preserving loss used for all experiments? What would be the effect of removing/applying it?
- How important is the structure-preserving loss for the domain translation experiments in Section 5.4?
- How does the work compare to existing trainable prior approaches?

---

> ### Author Response · Authors · 2024-11-29
>
> We sincerely appreciate the reviewers’ detailed and thoughtful feedback, which has been invaluable in improving the quality and clarity of our work. Below, we respectfully address the key concerns raised, highlighting the changes we have made to the manuscript in response to your suggestions.
>
> ## Reviewer Concerns and Clarifications
>
> ### 1. Missing Discussion & Contextualization of Related Work
> - **We deeply apologize** for the oversight in not properly citing and contextualizing the relevant prior work. We are extremely grateful that you brought this to our attention and have updated the manuscript to address your concerns:
>   - **Clarification of Our Contribution**
>     We have carefully refined our motivation to emphasize that our work extends the Gromov-Wasserstein framework by incorporating a **semantic metric space** when computing the GW cost. In contrast, prior works have primarily utilized the Euclidean space. This distinction helps highlight the novelty and importance of our approach.
>   - **Empirical Evidence**
>     In **Experiments 5.3 and 5.4**, we now provide empirical evidence demonstrating that using the semantic metric space leads to improved performance compared to prior works relying on the Euclidean space. These results validate the effectiveness of our proposed method and clarify its advantages.
>   - **Additional Intuition**
>     To further aid understanding, we have added additional intuitive explanations in **Appendix F** to illustrate why the semantic space offers advantages over the Euclidean space in this context. We hope this addition addresses your concerns and enhances the clarity of our work.
>
> ### 2. Experimental Section
> - **Clarification of the Structure-Preserving Loss Usage:**
>   We sincerely apologize for the lack of clarity in Sections 5.1 and 5.2 regarding the use of the structure-preserving loss. To resolve this, we have explicitly updated the experimental descriptions to indicate whether and which GW loss was employed in each experiment. We hope this revision provides the clarity needed to better understand the experimental setups.
>
> - **Comparison to Competing Methods with Flexible Priors:**
>   Thank you for the insightful suggestion to compare our score-based prior with other trainable priors. In response, we have included comparisons with methods leveraging flexible priors, such as Mixture of Gaussians (MoG) and VampPrior, in **Section 5.1**. Additionally, we now include comparisons with Latent Score-Based Generative Models (LSGM), demonstrating that our score-based models, combined with the SFS techniques, generally outperform these methods. We hope these results provide a more comprehensive understanding of the performance and advantages of our proposed approach.
>
> - **Quantitative Results for Section 5.4:**
>   We greatly appreciate your suggestion to include quantitative results for **Section 5.4**. In response, we have revisited the domain translation experiments on the CelebA dataset and added quantitative results to complement the qualitative findings. Specifically, we now report metrics quantifying the degree of semantic preservation during hair attribute translation. These results validate our claims with robust quantitative evidence and further demonstrate the effectiveness of our method.
>
> ---
>
> We are deeply grateful for the reviewers’ feedback, which has been instrumental in guiding us toward meaningful improvements. Your suggestions have significantly strengthened the clarity, rigor, and completeness of our manuscript. We respectfully request that you review our updated work, as we believe it now better addresses your concerns and more effectively communicates our contributions.
>
> Thank you once again for your time, effort, and invaluable insights. We truly appreciate your dedication to helping us refine and improve our work.
>
> [1] Vahdat, A., Kreis, K., and Kautz, J. (2021). Score-based Generative Modeling in Latent Space. arXiv preprint, arXiv:2106.05931. Retrieved from https://arxiv.org/abs/2106.05931.
>
> [2] Tomczak, J. M., and Welling, M. (2018). VAE with a VampPrior. arXiv preprint, arXiv:1705.07120. Retrieved from https://arxiv.org/abs/1705.07120.

---

> > ### Comment · Reviewer_HyiC · 2024-12-01
> >
> > While I appreciate the authors' extensive response and substantial revisions to better position the work within existing literature, the changes made to the manuscript effectively transform its core presentation and contributions. Such significant modifications, while appropriate, suggest the need for a fresh submission that can properly develop these ideas from the ground up.
> >
> > There remain several incomplete aspects that would need to be addressed, including e.g. :
> >
> > • The experimental section, while expanded to include comparisons with flexible priors, would benefit from more comprehensive comparisons with state-of-the-art methods. This would help establish the method's competitive advantages in current research contexts.
> >
> > • The addition of quantitative metrics for domain translation, while a step in the right direction, would benefit from better justification for the chosen evaluation metrics. These are not common evaluation metrics and it would be more convincing to base this experiment on existing quantitative evaluation schemes.
> >
> > • The use of semantic metric spaces over Euclidean spaces, while an interesting direction, would benefit from stronger theoretical foundations explaining their advantages.
> >
> > Given the substantial changes made to the paper, I recommend developing these aspects more fully in a revised submission that can properly establish and validate these contributions. The current framework shows promise, but would be better served by a fresh presentation that can thoroughly address these points while building on the improved positioning within the literature.

---

> ### Author Response · Authors · 2024-12-02
>
> We sincerely thank the reviewer for their thoughtful and thorough evaluation of our work, as well as their valuable feedback on the revised manuscript. We deeply appreciate the time and effort the reviewer has dedicated to carefully assessing our revisions and providing constructive suggestions for further improvement.
>
> If the reviewer feels that the revisions we have made have resulted in meaningful improvements to the manuscript, we kindly ask if they would consider reflecting this in their assessment score. However, we fully understand if the reviewer believes this is unwarranted at this stage. Regardless, we are truly grateful for the reviewer’s detailed insights.

---

### Official Review · Reviewer_KBQr · 2024-11-04

**Soundness:** 2
**Presentation:** 1
**Contribution:** 2
**Rating:** 3
**Confidence:** 3

**Summary:**

The following paper proposes score-based priors for VAEs in combination with Gromov-Wasserstein distance regularization for the domain adaptation problem.

**Strengths:**

The combination of score-based priors for VAEs with Gromov-Wasserstein distance seems to be a novel approach.

**Weaknesses:**

The contribution of the paper is unclear, making it difficult to follow. The authors frequently shift between topics in introduction section. Initially stating that they are addressing the distribution matching problem. However, they then build their motivation around fairness, robustness, causality, and explainability concepts, before ultimately changing the narrative, and  focus on the domain adaptation problem.

In addition, the evaluation provided is very poor, the central claims of the paper about "flexibility" are not supported. The evaluation is limited to domain adaptation, and the datasets used for comparison are simple. In addition, the paper proposes the use of structural regularization based on the Gromov-Wasserstein (GW) distance, but fails to provide any evaluation showing that this structural regularization is beneficial. Furthermore, the paper does not consider any datasets where the importance of structural regularization would be relevant.

Finally, the presentation can be improved. More examples and comparison on the face translation task, improved fonts on MNIST figure, comparison on at least, classic domain adaptation tasks as MNIST->SVHN, MNITS-MNIST-M. GW abbreviation is defined a few times in each section.

**Questions:**

1) Why are "flexible" representations important for trustworthiness, and what is your contribution? How do you demonstrate this flexibility?

2) Why are score-based priors useful for domain adaptation?

3) In the introduction, the authors criticize optimal transport methods with Euclidean cost functions. But the comparison with these methods in the domain adaptation task is missing, see papers (1,2,3).

4) Experimental settings in section 5.1 are unclear, why did the authors consider this dataset and not the more popular Celeb-A benchmark?

5) Why is it important to use structural regularization for the MNIST to USPS dataset? it would be more valuable to consider a typical dataset where Gromov-Wassesrstein regularization is applied. It is important to show how the method performs compared to other domain adaptation methods based on GW (4,5).

**References:**
1) https://openaccess.thecvf.com/content_CVPR_2020/papers/Li_Enhanced_Transport_Distance_for_Unsupervised_Domain_Adaptation_CVPR_2020_paper.pdf
2) http://proceedings.mlr.press/v139/fatras21a/fatras21a.pdf
3) https://proceedings.neurips.cc/paper_files/paper/2020/file/9719a00ed0c5709d80dfef33795dcef3-Paper.pdf
4) https://arxiv.org/pdf/2205.10738
5) https://arxiv.org/pdf/2303.05978

---

> ### Author Response · Authors · 2024-11-29
>
> We appreciate the reviewer's detailed feedback and believe that the points below address the key concerns.
>
> Reviewer Concerns and Clarifications
> 1. **Clarification of the contribution.**
>     We acknowledge that the introduction might appear to lack focus.
>     We start with the broad applicability of distribution matching (DM) methods, including fairness, robustness, causality, and explainability. We then proposed a new DM method inspired by previous DM works, and compared the performances between previous DM methods in the scope narrowed down to domain adaptation, translation, and fairness in the experiment section.
>     To improve clarity, we streamlined the introduction in the revised version to state the goal of the experiment section more clearly. Please be so kind to reread the **introduction section** and **methodolgy section** of are new manuscript.
>
> 2. **Response to Q1:**
>
>      We use the term "flexibility" to indicate the ability of the prior model to learn more complex distributions. We first mention that prior methods lack flexibility in their prior distribution or have expressive priors that are computational expensive and then propose to create a more flexible prior by using the score-based prior. We then define such a flexible model in **section 3.2** and provide specific synthetic dataset experiments to show the necessity of a flexible prior model in **section 5.1**.
>      We also make a connection to the term 'structure preserving' (SP) with the Gromov-Wasserstein (GW) distance in **section 3.1**. We explicitly show the benefits of introducing SP in the domain adaptation tasks in Table 1 as well. However, the original submission lacked SP comparisons in the experiments on translation. To give stronger evidence for SP, we explicitly added SP comparisons in the domain translation and domain adaptation tasks in the revised version.
>
> 3. **Response to Q2:**
>
>      We thank the reviewer for their insightful question. Prior works have shown that preserving the geometric properties of data is crucial for learning meaningful and robust representations [3, 4, 5]. Recent studies [1, 2] have demonstrated that enforcing geometry preservation constraints can induce disentanglement in the latent space. We believe these traits are particularly important for domain adaptation, as previous research has shown that disentangled representations lead to better performance in domain adaptation and translation tasks [6, 7, 8, 9].
>      However, past approaches face significant practical challenges. The simultaneous goals of preserving data geometry while matching a simple prior often lead to distortions within the latent space. To address this, [2] advocates for the use of more expressive priors, such as meta-priors, Gaussian mixtures, and neural priors, which provide greater flexibility in capturing complex data distributions while preserving geometric consistency. Despite their advantages, these learnable priors often have practical limitations, including poor scalability to high-dimensional spaces, significant computational expense, and instability during training.
>      Our method addresses these challenges by learning an expressive prior through a score-based model, leveraging the expressivity of diffusion models while ensuring stable performance. This stability is achieved by avoiding the computation of the UNET Jacobian term during updates to the encoder and decoder, significantly improving both scalability and training stability. We have also updated our paper to reflect this additional information.

---

> ### Author Response · Authors · 2024-11-29
>
> 4. **Response to Q3:**
>
>      We appreciate the opportunity to clarify how we define and compute our semantic-preserving loss.
>      We define our semantic-preserving loss as being calculated between the data dimension and the embedding dimension. For instance, in a domain adaptation setup with two domains—source and target—there will be one structural loss for each domain, namely $L_{SP}^{src}$ and $L_{SP}^{tgt}$. Specifically, $L_{SP}^{src}$ is computed between $x{src}$ and $z_{src} \sim q_{\theta}(z|x_{src})$, and similarly for $L_{SP}^{tgt}$.
>      On the other hand, Papers 1–5 that you kindly reminded us on, focus on formulating the loss in the embedding space between domains. Using the same domain adaptation example, there would be only one loss defined, $L_{OT/GW} = OT/GW(z_{src}, z_{tgt})$. Therefore, it is not meaningful to make a direct comparison between our structural loss and the losses proposed in these papers, as they are formulated differently and operate at distinct levels of abstraction.
>
> 5. **Response to Q4:**
>
>      In response to the reviewer’s suggestion, and after careful consideration, we agree with the proposed change. As such, we have decided to replace our FairFace dataset experiment with a CelebA benchmark. Specifically, we now focus on domain translation tasks between females with black hair and females with blonde hair. These results can be observed in **Section 5.4**.
>
> 6. **Response to Q5:**
>
>      We apologize for not providing sufficient motivation and clarity regarding the use of Gromov-Wasserstein (GW) regularization. To address this, we emphasize that studies [1, 2] have shown that enforcing geometry preservation constraints, such as GW distance, induces disentanglement in the latent space, which is critical for effective domain adaptation. In our revised **Section 3.3**, we further clarify that computing GW in a semantic space, rather than solely in Euclidean space as proposed by [1, 2], leads to more meaningful latent representations, as intuitively shown in **Appendix F**. Additionally, in the updated Section 5.3, we provide an empirical comparison of three configurations: no GW constraint, GW in Euclidean space (GW-EP), and GW in semantic space (GW-SP). These results demonstrate that GW-SP consistently outperforms the alternatives, reinforcing its effectiveness.
>
>
> [1] Uscidda, T., Eyring, L., Roth, K., Theis, F., Akata, Z., and Cuturi, M. (2024). Disentangled Representation Learning with the Gromov-Monge Gap. arXiv preprint, arXiv:2407.07829. Retrieved from https://arxiv.org/abs/2407.07829.
>
> [2] Nakagawa, N., Togo, R., Ogawa, T., and Haseyama, M. (2023). Gromov-Wasserstein Autoencoders. arXiv preprint, arXiv:2209.07007. Retrieved from https://arxiv.org/abs/2209.07007.
>
> [3] Chen, N., Klushyn, A., Ferroni, F., Bayer, J., and van der Smagt, P. (2020). Learning Flat Latent Manifolds with VAEs. arXiv preprint, arXiv:2002.04881. Retrieved from https://arxiv.org/abs/2002.04881.
>
> [4] Uscidda, T., Eyring, L., Roth, K., Theis, F., Akata, Z., and Cuturi, M. (2024). Disentangled Representation Learning with the Gromov-Monge Gap. arXiv preprint, arXiv:2407.07829. Retrieved from https://arxiv.org/abs/2407.07829.
>
> [5] Hahm, J., Lee, J., Kim, S., and Lee, J. (2024). Isometric Representation Learning for Disentangled Latent Space of Diffusion Models. arXiv preprint, arXiv:2407.11451. Retrieved from https://arxiv.org/abs/2407.11451.
>
> [6] Cai, R., Li, Z., Wei, P., Qiao, J., Zhang, K., and Hao, Z. (2019). Learning Disentangled Semantic Representation for Domain Adaptation. In Proceedings of the Twenty-Eighth International Joint Conference on Artificial Intelligence (IJCAI-2019), 2060–2066. Retrieved from http://dx.doi.org/10.24963/ijcai.2019/285.
>
> [7] Lee, S., Cho, S., and Im, S. (2021). DRANet: Disentangling Representation and Adaptation Networks for Unsupervised Cross-Domain Adaptation. arXiv preprint, arXiv:2103.13447. Retrieved from https://arxiv.org/abs/2103.13447.
>
> [8] Liu, Y.-C., Yeh, Y.-Y., Fu, T.-C., Wang, S.-D., Chiu, W.-C., and Wang, Y.-C. F. (2018). Detach and Adapt: Learning Cross-Domain Disentangled Deep Representation. In Proceedings of the IEEE Conference on Computer Vision and Pattern Recognition (CVPR), June 2018.
>
> [9] Bousmalis, K., Trigeorgis, G., Silberman, N., Krishnan, D., and Erhan, D. (2016). Domain Separation Networks. CoRR, arXiv:1608.06019. Retrieved from http://arxiv.org/abs/1608.06019.

---

### Author Response · Authors · 2024-11-29
**General Response 3**

## Conclusion

In its initial form, our paper admittedly fell short in several critical aspects, as pointed out by the reviewers. These shortcomings included unclear motivations, insufficient comparisons, and a lack of theoretical depth, particularly in the context of our proposed SFS trick and our Gromov-Wasserstein Semantic Preserving (GW-SP) loss. However, due to the extensive feedback and insightful critiques from the reviewers, we have undertaken a significant revamp of the manuscript. This includes major changes to the storyline, substantial experimental additions, and the incorporation of rigorous theoretical analysis. Most notably, the discovery of the connection between the SFS trick and the LSGM loss—thanks to the reviewers’ comments—has provided deeper insights and theoretical grounding. We believe these improvements make our work a meaningful contribution to the field.

We sincerely thank the reviewers for their invaluable feedback, which has enabled us to significantly improve the quality and clarity of the paper. We kindly request the reviewers to reread our substantially revised manuscript, as we believe it now addresses the raised concerns and offers meaningful contributions to the research community. We are deeply grateful for the reviewers’ guidance throughout this process, which has allowed us to refine our work and strengthen its potential impact.

[1] Roeder, G., Wu, Y., and Duvenaud, D. (2017). Sticking the Landing: Simple, Lower-Variance Gradient Estimators for Variational Inference. arXiv preprint, arXiv:1703.09194. Retrieved from https://arxiv.org/abs/1703.09194.

[2] Vahdat, A., Kreis, K., and Kautz, J. (2021). Score-based Generative Modeling in Latent Space. arXiv preprint, arXiv:2106.05931. Retrieved from https://arxiv.org/abs/2106.05931.

[3] Poole, B., Jain, A., Barron, J. T., and Mildenhall, B. (2022). DreamFusion: Text-to-3D using 2D Diffusion. arXiv preprint, arXiv:2209.14988. Retrieved from https://arxiv.org/abs/2209.14988.

[4] Nakagawa, N., Togo, R., Ogawa, T., and Haseyama, M. (2023). Gromov-Wasserstein Autoencoders. arXiv preprint, arXiv:2209.07007. Retrieved from https://arxiv.org/abs/2209.07007.

[5] Uscidda, T., Eyring, L., Roth, K., Theis, F., Akata, Z., and Cuturi, M. (2024). Disentangled Representation Learning with the Gromov-Monge Gap. arXiv preprint, arXiv:2407.07829. Retrieved from https://arxiv.org/abs/2407.07829.

[6] Tomczak, J. M., and Welling, M. (2018). VAE with a VampPrior. arXiv preprint, arXiv:1705.07120. Retrieved from https://arxiv.org/abs/1705.07120.

[7] Kladny, K.-R., von Kügelgen, J., Schölkopf, B., and Muehlebach, M. (2024). Deep Backtracking Counterfactuals for Causally Compliant Explanations. arXiv preprint, arXiv:2310.07665. Retrieved from https://arxiv.org/abs/2310.07665.

[8] Pan, Y., and Bareinboim, E. (2024). Counterfactual Image Editing. arXiv preprint, arXiv:2403.09683. Retrieved from https://arxiv.org/abs/2403.09683.

[9] Komanduri, A., Zhao, C., Chen, F., and Wu, X. (2024). Causal Diffusion Autoencoders: Toward Counterfactual Generation via Diffusion Probabilistic Models. arXiv preprint, arXiv:2404.17735. Retrieved from https://arxiv.org/abs/2404.17735.

[10] Sauer, A., and Geiger, A. (2021). Counterfactual Generative Networks. arXiv preprint, arXiv:2101.06046. Retrieved from https://arxiv.org/abs/2101.06046.

---

### Author Response · Authors · 2024-11-29
**General Response 2**

## Revisions and Improvements to Experiments

All replaced experiments have been moved to the appendix.

- We acknowledge the valid concerns raised by **Reviewer KBQr**, **Reviewer HyiC**, and **Reviewer 7J8h** regarding the lack of clear evidence supporting the benefits of our learnable, expressive prior. To address this, we conducted synthetic experiments comparing different learnable priors with conventional priors. The results of these experiments are presented in **Section 5.1**, with additional detailed information and results provided in **Appendix H.2** (pages 25–29).
  **Note:** We would like to apologize for forgetting to highlight the text changes in **Section 5.1**. We want to assure the reviewers that the corresponding changes have been made, and we sincerely regret this oversight. Thank you for your understanding.

- As discussed above, and with the valuable suggestions from **Reviewer woTo** and **Reviewer R3FP** in mind, we have conducted stability experiments comparing LSGM and the SFS trick. Specifically, we measured stability when training on low noise levels, demonstrating that our model is significantly more stable under these conditions. This stability enables more precise modeling of the posterior distribution observed in their negative log-likelihood values, as detailed in **Section 3.2.1**.

- We would like to address the concerns regarding the weaknesses in our domain translation experiment, as highlighted by **Reviewer KBQr** and **Reviewer woTo**. Specifically, these concerns pertain to the effectiveness of the translation when evaluated qualitatively and the lack of supporting quantitative metrics. In response, and as suggested by **Reviewer KBQr**, we have conducted additional experiments on the CelebA dataset, focusing on image translation between black-haired females and blonde-haired females. To provide quantitative support, we utilized image retrieval as our evaluation metric. Qualitatively, we demonstrate that our approach not only preserves the semantic content of the input images but also successfully translates them to the target domain. These improvements are detailed in **Section 5.4**.

- **Reviewer KBQr**, **Reviewer HyiC**, **Reviewer woTo**, and **Reviewer 7J8h** raised important concerns regarding the GW metric and questioned the lack of comparisons between different GW metric configurations—specifically, the absence of GW regularization, GW in the pixel/Euclidean space, and GW in the semantic space. In response, we have reran all experiments with these three configurations to demonstrate the significant improvement in efficacy when using the semantic space.

  We argue that for high-dimensional, real-world image datasets, it is more appropriate to perform comparisons in the semantic space rather than the pixel space. To further clarify the drastic improvements observed in certain tasks (e.g., CelebA) and the more modest improvements in others (e.g., domain adaptation), we present AUROC scores in **Appendix F**. The AUROC scores measure how well a model can separate two classes (e.g., positive and negative) or distinguish between two distributions (e.g., source and target domains). These scores, computed on MNIST, USPS, and CelebA datasets in both the pixel and semantic spaces, provide additional evidence supporting our claim that the semantic space is more appropriate for high-dimensional images.

- We would also like to respectfully point the reviewers to the additional experiments in the appendix that may have been overlooked during the review process.
  - In **Appendix D**, we showcase the effectiveness of our model in achieving strong domain adaptation results in a limited source label setting. Specifically, our model requires less than 1% of the original source labels to achieve accurate predictions on the target labels. This claim is further supported by UMAP visualizations of the domain adaptation classes, which clearly show well-separated clusters in the latent space.
  - Furthermore, in **Appendix E**, we present results on domain translation in a multi-domain setting using rotated MNIST datasets with three different angles (0°, 30°, 60°). Our method demonstrates impressive performance in translating between these angles while preserving both stylistic and semantic information. Although this task may appear trivial, it is, in fact, a challenging problem to manipulate MNIST images without altering their semantics, a topic closely related to ongoing research in causal-based image generation \[6, 7, 8, 9, 10\].

  We hope these additional results provide a more comprehensive understanding of our model's capabilities and its potential impact in these areas.

---

### Author Response · Authors · 2024-11-29
**General Response 1**

First and foremost, we would like to extend our apologies to the reviewers for the delayed submission of our rebuttals. We recognize that our initial submission did not meet the high standards expected by the reviewers, and we take full responsibility for this shortcoming. Our intention was to dedicate as much time as possible to addressing these issues by refining our storyline, deepening our insights, and conducting additional experiments, as we felt our initial submission was inadequate.

We are deeply grateful to all the reviewers for their thoughtful and constructive critiques. Their insights have been instrumental in guiding us toward significant improvements in the quality and clarity of our work. We are especially thankful to the reviewers for pointing us toward relevant literature, enabling us to deepen our understanding of related works. We have carefully reviewed all the suggested papers and humbly acknowledge that we were previously unaware of the paper highlighted by **Reviewer HyiC** and **Reviewer woTo** related to current Gromov-Wasserstein-based methods. In light of this, we have made corresponding changes to the storyline of the paper. This guidance has been invaluable in broadening our perspective and situating our work within the existing body of research. We sincerely appreciate the reviewers’ time and effort in evaluating our submission and for their role in helping us grow through this process.

## Substantial Changes to the Storyline

- Thanks to **Reviewer KBQr** and **Reviewer 7J8h**, we have completely revamped the introduction and overall storyline by adding clearer motivations and decoupling our novelty from past works. We have also made the focus more precise, as our revisions have made the narrative more cohesive and focused.
- We are especially grateful to **Reviewer HyiC** and **Reviewer woTo** for bringing to our attention prior works that have also utilized GW metrics within their frameworks. In response, we have ensured that these works are appropriately credited in our methodology section. Furthermore, we have revised the storyline to better highlight our contributions.
- **Reviewer 7J8h** posed important and deeply critical questions that helped us recognize that some of our wording suggested a level of certainty that was not fully justified by the evidence provided. In response, we have carefully revised our claims, adopting a more measured tone and ensuring all claims are substantiated with appropriate evidence and references to related works.

## Significant Revisions to the Methodology Section

- We would like to extend our special thanks to **Reviewer HyiC** for providing the invaluable insight of connecting the Score Function Substitution (SFS) trick to Latent Score-Based Generative Models (LSGM) \[2\] in a manner resembling the Score Distillation Sampling (SDS) loss \[3\]. Initially, the derivation of the SFS trick appeared relatively straightforward, and its effectiveness was not immediately evident as suggested by **Reviewer R3FP**. However, this insight allowed us to rigorously prove that SFS is, in fact, a distilled version of the LSGM loss. Specifically, by applying the Sticking-the-Landing principle \[1\] to the LSGM objective, we demonstrate that a loss proportional to the SFS trick can be derived (the full proof can be seen in **Appendix E.1**).

  This discovery provides theoretical grounding for the stability of our loss function at lower noise levels—due to the removal of the Jacobian term in LSGM, which serves as a control variate—and highlights practical advantages, such as reduced memory usage. We further corroborate this statement by conducting a stability analysis between LSGM and SDS in **Section 3.2.1**, showcasing the empirical benefits of our approach.

  This finding reveals that the SFS trick is significantly more novel and impactful than we initially understood, and we believe it could pave the way for future advancements in latent score-based/diffusion generative models.

- As importantly highlighted by **Reviewer HyiC** and **Reviewer woTo**, prior works, such as GWAE \[4\], have also adopted similar GW metrics. In response, we have clarified in our methodology section that our work builds upon their GW metric and extends its application by exploring its use in the semantic space for high-dimensional image data. This adjustment ensures proper acknowledgment of prior contributions while emphasizing our novel extensions.

---

### Author Response · Authors · 2024-12-04
**Final Remark and Request for Consideration**

We sincerely thank the reviewers for their detailed feedback and the time they have invested in evaluating our work. Their critiques have greatly enhanced the clarity, rigor, and overall quality of our paper, and we deeply respect their thoughtful assessments. However, we would like to humbly request reconsideration of our submission in light of the revisions and responses we have provided.

While some feedback suggested that the storyline of our paper changed significantly, we respectfully clarify that the core storyline remains the same. Our revisions primarily focused on refining and narrowing the narrative to make it more specific and coherent. For instance, the semantic-space Gromov-Wasserstein loss has always been a central component of our work, and our revisions primarily involved acknowledging prior works we were previously unaware of and presenting our loss as an adaptation. This adjustment ensures proper credit while maintaining the novelty and utility of our approach.

Similarly, the deeper insight we gained into the Score Function Substitution (SFS) trick has only strengthened our contribution without altering its essence. Moreover, we augmented our work with additional experiments to substantiate our claims and extensively cited related literature to further validate our methodology and contextualize our contributions.

Regarding concerns about state-of-the-art (SOTA) comparisons, we would like to respectfully emphasize that the primary goal of our work is not to establish new SOTA benchmarks. Instead, our focus is on introducing a very adaptable and general framework for distribution matching with a flexible prior, demonstrating its versatility across diverse downstream tasks. With promising initial performance, we believe this broader contribution provides a strong foundation for future advancements in the field, which could potentially lead to SOTA results in downstream tasks as new techniques and methodologies are developed on top of our framework.

Once again, we are deeply grateful for the reviewers’ dedication and constructive feedback, which have been invaluable throughout this process. We sincerely hope that the reviewers will reconsider our revised submission, as we believe it addresses their concerns and presents meaningful contributions to the research community. Thank you for your time, effort, and commitment to fostering impactful research.

---

### Meta-Review · Area_Chair_pW9J · 2024-12-21

**Metareview:**

The paper presents a distribution matching approach using score-based priors and Gromov-Wasserstein (GW) distance regularization. Reviewers acknowledge the technical soundness and potential contributions of the work. Still, they express major concerns about insufficient acknowledgment of prior work, limited experimental validation (ablation studies, comparisons with existing methods, and quantitative results), and inadequate justification for the benefits of score-based priors and structural preservation. The authors' rebuttal provides additional experiments and expanded comparisons, addressing some of the important concerns. However, reviewers find that the paper requires substantial modifications to integrate the experiments and clarifications presented during the rebuttal. As such, the AC recommends further improvements before the paper can be accepted.

**Additional Comments On Reviewer Discussion:**

Overall, the authors make significant efforts to address reviewers’ concerns during rebuttal. However, some concerns remain unresolved, and substantial changes are needed to integrate the clarifications and experiments into the paper.

Reviewer KBQr expressed major concern about the unclear contribution and experiment designs regarding baselines and datasets. The authors provided clearer clarifications and
more evaluation results on CelebA as requested. Without further engagement from the reviewer, these concerns are carefully considered but less weighed.

Reviewer HyiC primarily mentioned the missing discussion and comparisons with related works. The authors added substantial experiment results correspondingly, which, as pointed out by the reviewer, requires significant modifications to the original paper.

Reviewer woTo pointed out a series of weaknesses regarding the contribution, experiment settings, and the significance of the experimental results. The authors carefully clarify the concerns in the rebuttal. Without further engagement from the reviewer, these concerns are carefully considered but less weighed.

Reviewer R3FP's major concern was the computational complexity of the score-based priors. While the authors emphasized that their method provides computational savings, the reviewer expresses further concerns in the following discussion that no time complexity analysis is performed.

Reviewer 7J8h showed concern about the lack of concrete evidence for claimed benefits and the use of simple datasets. This reviewer remained unconvinced by the explanation and experiment results added by the authors.

Given the above unaddressed concerns and substantial changes need to be made, the AC recommends further improvements before the paper can be accepted.

---

### Decision · Program_Chairs · 2025-01-22

Reject